# *Pasteurellosis* Vaccine Commercialization: Physiochemical Factors for Optimum Production

Siti Nur Hazwani Oslan [1,2,*], Joo Shun Tan [3], Abdul Hafidz Yusoff [4], Ahmad Ziad Sulaiman [4], Mohd Azrie Awang [1,*], Azwan Mat Lazim [5], Si Jie Lim [6,7], Siti Nurbaya Oslan [6,7,8] and Mohd Zamri Saad [9] and Arbakariya B. Ariff [2]

1. Faculty of Food Science and Nutrition, Universiti Malaysia Sabah, Jalan UMS, Kota Kinabalu 88400, Sabah, Malaysia
2. Department of Bioprocess Technology, Faculty of Biotechnology and Biomolecular Sciences, Universiti Putra Malaysia, Serdang 43400, Selangor, Malaysia; arbarif@upm.edu.my
3. Bioprocess Technology, School of Industrial Technology, Universiti Sains Malaysia, Gelugor 11800, Pulau Pinang, Malaysia; jooshun@usm.my
4. Gold Rare Earth and Material Technopreneurship Centre (GREAT), Faculty of Bioengineering and Technology (FBET), Universiti Malaysia Kelantan Kampus Jeli, Locked Bag 100, Jeli 17600, Kelantan, Malaysia; hafidz.y@umk.edu.my (A.H.Y.); ziad@umk.edu.my (A.Z.S.)
5. Department of Chemical, Faculty of Science and Technology, Universiti Kebangsaan Malaysia, Bangi 43600, Selangor, Malaysia; azwanlazim@ukm.edu.my
6. Enzyme and Microbial Technology Research Centre, Centre of Excellence, Universiti Putra Malaysia, Serdang 43400, Selangor, Malaysia; gs58349@student.upm.edu.my (S.J.L.); snurbayaoslan@upm.edu.my (S.N.O.)
7. Enzyme Technology and X-ray Crystallography Laboratory, VacBio 5, Institute of Bioscience, Universiti Putra Malaysia, Serdang 43400, Selangor, Malaysia
8. Department of Biochemistry, Faculty of Biotechnology and Biomolecular Sciences, Universiti Putra Malaysia, Serdang 43400, Selangor, Malaysia
9. Research Centre for Ruminant Diseases, Faculty of Veterinary Medicine, Universiti Putra Malaysia, Serdang 43400, Selangor, Malaysia; mzamri@upm.edu.my
* Correspondence: snhazwanioslan@ums.edu.my (S.N.H.O.); ma.awang@ums.edu.my (M.A.A.)

**Abstract:** *Pasteurella* spp. are Gram-negative facultative bacteria that cause severe economic and animal losses. *Pasteurella*-based vaccines are the most promising solution for controlling *Pasteurella* spp. outbreaks. Remarkably, insufficient biomass cultivation (low cell viability and productivity) and lack of knowledge about the cultivation process have impacted the bulk production of animal vaccines. Bioprocess optimization in the shake flask and bioreactor is required to improve process efficiency while lowering production costs. However, its state of the art is limited in providing insights on its biomass upscaling, preventing a cost-effective vaccine with mass-produced bacteria from being developed. In general, in the optimum cultivation of *Pasteurella* spp., production factors such as pH (6.0–8.2), agitation speed (90–500 rpm), and temperature (35–40 °C) are used to improve production yield. Hence, this review discusses the production strategy of *Pasteurella* and *Mannheimia* species that can potentially be used in the vaccines for controlling *pasteurellosis*. The physicochemical factors related to operational parameter process conditions from a bioprocess engineering perspective that maximize yields with minimized production cost are also covered, with the expectation of facilitating the commercialization process.

**Keywords:** pasteurellosis; *Pasteurella multocida*; *Mannheimia haemolytica*; cultivation process; physio-chemical factors; maximizing yields; minimizing cost

## 1. Introduction

*Pasteurellaceae* family bacteria, which are Gram-negative, can cause infectious diseases in domestic and wild animals that are endemic and epidemic in a wide population area [1,2].

The bacteria are responsible for considerable economic losses in livestock worldwide. This causative agent is responsible for 30% of the total cattle deaths around the world; North America alone has reported losses of one billion dollars annually in this industry [3]. In addition, it is responsible for considerable economic losses in the pork and poultry industries worldwide. The *Pasteurellaceae* family members associated with animal diseases are *Pasteurella multocida, P. trehalosi*, and *Mannheimia haemolytica* [4]. These bacteria are potential pathogens that are able to infect buffaloes, cattle, goats, sheep, poultry, pigs, avian species, and others [5]. Katechakis et al. [6] have reported a case in which a lady who lived with cats was infected by *P. multocida*, without any evidence of animal bites or scratches, but only by kissing her pets and touching them with her mouth. *Pasteurellosis* outbreaks in Malaysia were widely recorded in the early 1970s to 1990s, particularly regarding clinical epidemics of hemorrhagic septicemia in cattle and buffaloes [7]. Strikingly, the prevalence rate of *pasteurellosis* ranges from 1.0% to 3.2% (2014 to 2016), based on cases received at the Veterinary Research Institute, Ipoh, Malaysia [7]. Furthermore, Khoo et al. [8] revealed that the *P. multocida* serogroup F is circulating in Malaysia and affects livestock clinically.

Furthermore, keeping livestock healthy to increase output is critical to the agricultural economy. The vaccination of animals against various infectious illnesses prevalent in different geographical locations is a fundamental measure of good husbandry practices that plays a significant role in preserving animal health and limiting economic losses owing to infectious diseases. Most vaccines that are used for the vaccination of animals or humans are either live or inactivated. Traditional veterinary vaccinations protect animals from the risks of numerous infectious illnesses. Vaccination boosts the immune system of the animal and prepares it to fight illnesses caused by harmful bacteria. It is the most successful method used to prevent the transmission and spread of animal disease outbreaks, resulting in complete security and public health. As the necessity of vaccination is repeatedly highlighted, research into the large-scale industrial manufacturing of vaccines for controlling *pasteurellosis* has become vital to meet the growing demand.

*Pasteurella* spp. are fastidious bacteria that require a rich medium because their ability to metabolize nutrients, such as sugars, amino acids, peptides, and vitamins are different according to strain [9]. However, certain *P. multocida* strains are frequently grown in brain–heart infusion (BHI), Terrific Broth (TB), tryptic soy broth (TSB), and serum broth [9,10]. Consequently, the components of the medium and culture conditions have a substantial impact on the generation of viable cells and biomass in *P. multocida*. As an alternative to traditional media, a novel medium for each strain or species might be developed, resulting in increased biomass output and economic benefits, such as cost and fermentation time reduction.

This review presents strategies in the production of *Pasteurella* spp. that can potentially be used as a vaccine for the controlling of *pasteurellosis*. The physicochemical factors related to the growth substrate and operational parameters process conditions are highlighted from a bioprocess engineering perspective, to maximize the yields of production prior to commercialization.

## 2. *Pasteurella multocida* and *Mannheimia haemolytica*

*Mannheimia haemolytica* is a member of the family *Pasteurellaceae* (formerly *Pasteurella haemolytica* biotype 1). Because different serotypes of *P. multocida* have distinct characteristics, they cause a variety of diseases in animals, including hemorrhagic septicemia (HS) in ruminants (serotypes B:2 or E:2), atrophic rhinitis in pigs (serogroups A and D), and fowl cholera in poultry (serotypes A:1, A:3, and A:4) [11,12]. Bovine respiratory disease (BRD), caused by *M. haemolytica*, is also a serious problem in the dairy and meat cattle sectors [13]. *P. multocida* is recognized as a typical microflora in most healthy animals' upper respiratory tracts (URT). *P. multocida* colonizes the mucosa of the URT and then infects the air sacs and lungs, as well as the nasopharynx, respiratory tract, and other organs of numerous animals [14]. Bacteria can enter the bloodstream from the mucosa and multiply in many organs, particularly the liver and spleen, via an unknown process

that is probably connected to movement in URT macrophages [2]. *P. multocida* may be verified microscopically, biochemically, and quickly using the Vitek2 compact system, an enzyme-linked immunosorbent test (ELISA), mouse lethality bioassay, and a polymerase chain reaction (PCR) [15]. In addition, matrix-assisted laser desorption/ionization time-of-flight mass spectrometry (MALDI-TOF) has been identified as a quick and dependable alternative method for differentiating most members of the *Pasteurellaceae* family [16].

Furthermore, *pasteurellosis* is a disorder that is frequently connected with stress situations, such as transportation or changes in climatic conditions, diet, and injuries [17,18]. Table 1 shows the illnesses produced by *Pasteurella* spp. in agricultural and domestic animals. *P. multocida*, on the other hand, is a zoonotic human pathogen that enters subcutaneous tissues through cat- and dog-bite wounds [2]. *Pasteurella* spp. was identified from about 50% and 75% of dog and cat attacks on humans alone, according to Talan et al. [19].

**Table 1.** Diseases caused by *Pasteurella* organisms in agricultural and domestic animals.

| Host Species | Disease | Microorganisms | References |
|---|---|---|---|
| Cattle/buffalo | Hemorrhagic septicemia (HS) | *P. multocida* serotype B:2, E:2 | [11,20,21] |
| Cattle | HS-like septicemic disease | *P. multocida* serotype B:2,4 | [22,23] |
| Cattle | Pneumonic *pasteurellosis* | *M. haemolytica* serotype A1 *P. haemolytica* A | [23,24] |
| Cattle | Bovine respiratory disease (BRD) | *P. multocida* A:3 *M. haemolytica* | [13,25] |
| Sheep and goats | Pneumonic *pasteurellosis*, Septicemic *pasteurellosis* | *P. haemolytica* A *P. trehalosi* *P. multocida* serotype B:2 | [26,27] |
| Pigs | Atrophic rhinitis | *P. multocida* serotype D | [28] |
| Poultry/turkeys | Fowl cholera | *P. multocida* serotype A:1, A:3, A:4 *P. multocida* type F in turkeys | [29,30] |
| Birds/ducks | Fowl cholera | *P. multocida* serotype A *P. multocida* serotype D | [10,31,32] |

Generally, *P. multocida* infection in livestock industries (ruminant for HS) is controlled by plain broth bacterins, alum precipitation, aluminium hydroxide gel, or oil-adjuvant vaccines. However, these vaccines have certain limitations, encompassing a shorter duration of immunity, the need for the animal to be restrained, and the swelling at the site of vaccine inoculation [3,33,34].

In recent years, molecular techniques have been used widely as approaches for developing effective vaccines to control *pasteurellosis* (Table 2). For example, the mutant *gdhA* derivative *P. multocida* B:2 has been developed as an attenuated live vaccine and its stability against HS disease has been proven [35]. A killed vaccine candidate has been studied by Arif et al. [36], wherein an outer membrane protein preparation via anti-idiotype *P. multocida* vaccine has proven that the outer membrane protein (OMPs-anti-idiotype) vaccine induced high levels of antibody titers compared to bacterin vaccines on protection studies in a rabbit model. In addition, the chitosan nanoparticle DNA vaccine based on the *ptfA* gene of *P. multocida* was shown to enhance the immune response to a *P. multocida* challenge in chickens [37].

**Table 2.** Discoveries regarding *P. multocida* vaccines in the past 10 years (2011–2021).

| Animals | Vaccine/Vaccine Targets | Challenge Strains | Immune Profiles [a] | Day [b] | N | % | References |
|---|---|---|---|---|---|---|---|
| | | | | | | Protection Efficacies | |
| Live-attenuated and Bacterin | | | | | | | |
| Calves | *M. haemolytica* serotype A1 and *P. multocida* | *M. haemolytica* serotype A6 | ↑ Abs | 21 | 15–17 | Reduced lung bacteriology | [38] |
| Rabbits | *P. multocida* B:2 | | ↑ Abs | 7 | 8 | 25.0 | [36] |
| Mice | *P. multocida* (alginate microparticle) | *P. multocida* | - | 7 | 5 | 6 log protection | [39] |
| Mice | *P. multocida* A strain PMSHI-9 (iron-inactivated; bDNA adjuvanted) | *P. multocida* | ↑↑ IL-6, IL-12, Abs | 28 | 10 | 100.0 | [40] |
| Ducks | *P. multocida* 0818 strain Δ*fur* | *P. multocida* 0818 strain | ↑ SI (serum IgY and bile IgA) | 10 | 50 | 62.0 | [41] |
| Antibody/Anti-idiotype | | | | | | | |
| Rabbits | *P. multocida* B:2 | | ↑ Abs | 7 | 8 | 100.0 | [36] |
| Recombinant | | | | | | | |
| Chickens | *ptfA* (subunit with Freund's complete adjuvant) | CVCC474 A:1 | ↑↑ SI, Abs, IFN-γ | 15 | 20 | 45.0 | [42] |
| Pigs | N-terminal of *P. multocida* toxin (N-PMT) | *P. multocida* | ↑↑ IgG | 28 | 9 | Lower turbinate atrophy level | [43] |
| Goats | *P. multocida* B:2 fimbrial protein | *P. multocida* B:2 | ↑↑ IgG and IgA | 3 | 3–6 | Zero/Reduced bacterial burden in organs | [44] |
| Turkeys | *P. multocida* P-1059 recombinant filamentous hemagglutinin peptide (rFHAB2) adjuvant | *P. multocida* P-1059 or χ73 | - | 8 | 40 | 70.0–75.0 | [20] |
| DNA Vaccine | | | | | | | |
| Chickens | *ptfA* (inulin-adjuvant) | CVCC474 A:1 | ↑↑ SI, IL-2, IFN-γ; ↑ IL-4; ≡ Abs | 15 | 20 | 55.0 | [45] |
| Chicken | *ptfA* (chitosan-adjuvant) | CVCC474 A:1 | ↑↑ SI, IL-2, IFN-γ; ≡ IL-4 | 15 | 25 | 68.0 | [37] |
| Mice | *tbpA* (IL-2-adjuvant) | *P. multocida* serotype B:2 (strain P52) | ↑↑ SI, IgG | 48 h | 12 | 66.6 | [46] |
| Chicken | *ompH*, *ompA* (divalent and fusion) | CVCC474 A:1 | ↑↑ SI, IFN-γ | 15 | 20 | 70.0–75.0 | [47] |

[a] Immune response elicited after vaccination and before challenge (↑↑, ↑, and ≡ are the indicator of significantly higher, higher, and not significant, respectively). [b] Experimental period after a challenge with virulent *P. multocida*. Abbreviations: Abs (antibodies); IL (interleukin); bDNA (bacterial DNA); SI (serum immunoglobulin); IFN (interferon).

Other recombinant vaccines, that are, the modified-live *M. haemolytica* and *P. multocida* vaccines, were injected into young dairy calves by Aubry et al. [48], demonstrating high effectiveness in increasing the titers of antibodies against *M. haemolytica*. The *fur* mutant strain of *P. multocida* demonstrated a 62% protection efficacy against severe lethal challenge with the wild-type *P. multocida*, after orally inoculating the ducks [41]. Other studies have also been conducted by the cross-protection of an *M. haemolytica* A1 *Lkt⁻/P. multocida hyaE* bovine respiratory disease vaccine against experimental challenge with *M. haemolytica* A6 on calves. The results showed high protection for calves after the subsequent challenge [38].

Although in vitro and in vivo vaccine discoveries are time-consuming and costly, besides possessing animal ethical issues, a recent in silico study has shed light on the search for serotype-independent vaccine candidates [49]. The reverse vaccinology strategy that was performed successfully elicited the most promising order of *P. multocida* PlpE epitopes in a multi-epitope model that can be used as a novel subunit vaccine candidate [49]. Therefore, it is important to explore continuously improvized vaccine design strategies that later require improved biomass and cell viability with effective control methods.

### 3. *Pasteurella* spp. Production

*3.1. Current Status and Challenges*

Recent demands for veterinary vaccines in the treatment of *pasteurellosis* disease have increased, as reflected by the applications for vaccination programs. Currently, research on production targeted at the commercialization of a *Pasteurella* vaccine is limited. Othman [50] reported that the cell concentrations of *P. multocida* B:2 deemed to be effective for vaccination in the host ranged from $10^5$ to $10^7$ CFU/mL. Therefore, the designs of physiochemical studies reviewed in this article allow wider knowledge distribution among researchers in the field. The large-scale production of veterinary vaccines can be conducted through the bacterial fermentation process. For example, a vaccine against a *P. multocida* B:2 mutant, which was produced via bacterial fermentation with improved reproducibility and scalable synthesis at lower production cost and energy input, was found to be suitable as a live vaccine candidate for controlling outbreaks of HS [9]. Nowadays, the cost of a commercially available culture medium used to grow bacteria is extremely high. For instance, the culture of *P. multocida* B:2 mutant using BHI broth and yeast dextrose broth (YDB) with a cell viability of $7.3 \times 10^9$ and $3.4 \times 10^8$ CFU/mL in a single operation of cell production (in US dollars) of USD 10.10 was significantly higher than that in an optimized medium YDB, USD 1.84/L [9]. Figure 1 shows that one example of *P. multocida* isolated from the lungs of an infected buffalo exhibited grey, non-mucoid, and non-hemolytic colonies on 5% sheep-blood agar after 24 h of incubation at 37 °C and showed no growth on MacConkey agar.

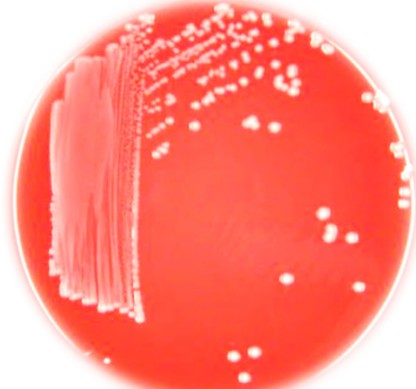

**Figure 1.** *P. multocida* B:2 colony on 5% (*v/v*) sheep-blood agar. The colony was observed after overnight incubation at 37 °C (unpublished data).

Therefore, lower costs for cultivation, as well as for scaling up, are important in vaccine manufacturing. The efficient development of a vaccine, especially in terms of

growth medium formulation, is necessary to obtain high bacterial cell viability and biomass. The first step of mass production is to identify the medium formulation with minimally acceptable costs for high-quality biomass production. Nevertheless, the medium must contain the basic requirements for cell biomass production and growth, supplying enough energy for biosynthesis and cell maintenance of the bacterial vaccine.

Notably, the usage of cheap nitrogen and carbon sources can drastically reduce the cost of production. Many studies have reported the increment of biomass and the cell viability of bacteria was presumably dependent upon the nutrients [51–53]. With this in view, nitrogen and carbon sources that are present in the medium can influence biomass bacterial production, whereby a limited supply of nutrient types and their concentrations in the medium can restrict the growth of microbial cells or product formation [54].

### 3.1.1. Shake Flasks

Shake flask fermentations are normally carried out in a laboratory and are widely used to investigate microbial culture behavior on a small scale, mainly to explore feasibility [55]. In bioprocess development applications, the flasks are mainly used for screening and early bioprocess development, which, in medium-level optimization and the preparation of starter cultures in shake flasks, are applied as a pre-culture of stirred-tank fermentation [56]. In addition, the shake flask system enables the performance of a high number of cost-efficient parallel fermentation procedures on a small scale within a set of optimum fermentation conditions [55,57]. The fermentation could be carried out in Erlenmeyer flasks, based on certain parameters such as minimal media, carbon sources, and precursors at specific pH levels that are incubated at certain temperatures and agitation rates for a certain period of time [57].

Moreover, nitrogen and carbon sources in the culture medium are important for biosynthesis and energy generation [58]. Oslan [52] reported a preliminary screening on cost-effective nitrogen and carbon sources that can present an alternative for the pricey commercial brain-heart infusion (BHI) medium—an enriched medium for *P. multocida* growth [10]. Optimization of the medium components has been conducted in a shake-flask study using the response surface methodology (RSM); subsequently, the central composite design (CCD) has been used to investigate the optimum value of the selected medium component to achieve the maximum biomass yield of mutant *P. multocida* B:2 [52].

In the fermentation process, optimization on the media using classical methods, the "one-factor-at-a-time" (OFAT) strategy, is extremely time-consuming, expensive because of its many variables, and needs a relatively large number and repetition of experiments with compromised accuracy [54]. However, using the RSM technique, which comprises modern mathematical/statistical techniques, leads to a more efficient, economical, robust as well as effective media optimization. The newly formulated medium was also successfully optimized and developed by RSM on a shake-flask scale. The yeast extract, glucose, sodium chloride, and sodium phosphate were proven to influence biomass production. The productivity of biomass production has been improved by 52.72% compared to the non-optimization of the medium [52].

### 3.1.2. Bioreactors

For biomass production on an industrial scale, a large working volume like a bioreactor is the most suitable method, whereby many restrictions encountered under shake-flask conditions can also be avoided. The cultivation of *Pasteurella* spp. can be conducted using either batch, fed-batch, or continuous-mode underfeeding strategies by using a bioreactor. Subsequently, numerous studies have demonstrated the use of these cultivation approaches for *Pasteurella* spp. [10,59–61].

The batch cultivation of *Pasteurella* spp. possesses a significant need to boost bacterial growth and, hence, the maximum cell density that can be produced [60,62]. Study of the growth profile of *Pasteurella* cells can also be preliminarily undertaken during the batch phase. In a controlled environment in a bioreactor under batch cultivation, the manipulation

of parameters (temperature, pH, and dissolved oxygen tension (DOT)) can further enhance its performance at the industrial scale. It is noteworthy that such a strategy has been conducted while producing the mutant *P. multocida* B:2 biomass in a 5 L bioreactor [52], and continuously, using a 2-L bioreactor supplemented with the amino acid, histidine [9].

In the meantime, the growth and product formation kinetics of the bovine pathogen *M. haemolytica* strain OVI-1 had also been investigated effectively in a continuous culture model [51]. The leukotoxin (LKT) concentration and the bacterial biomass were enhanced by supplementing a carbon-limited medium with an amino acid mixture or a mixture of cysteine and glutamine. Following that study, there has been no report on the fed-batch cultivation mode of *Pasteurella* spp. Table 3 shows the *Pasteurella* spp. cultivations studied in batch and continuous modes, with the improved yields of different targeted products.

**Table 3.** General fermentation conditions of *Pasteurella* species.

| Strain | Optimal Growth Condition | | Agitation Speed (rpm) | References |
|---|---|---|---|---|
| | Temperature (°C) | pH | | |
| *P. multocida* B:2 | 35–40 | 6–8 | 500–550 | [53,61] |
| *gdhA* derivative *P. multocida* B:2 | 37 | 6–7 | 200–700 | [9] |
| *aroA* derivative *P. multocida* B:2 | 37 | 7.4 | 200 | [20] |
| *P. multocida* A | 37 | 7.2 | 90–200 | [59] |
| *P. multocida* | 37 | 7.2 | 200 | [63] |
| *P. multocida* | 35–40 | 7.2–8.2 | 500–1000 | [10] |
| *M. haemolytica* | 37 | 7.0 | 400–550 | [64] |
| *M. haemolytica* | 35–37 | 7 | 200 | [65] |
| *P. septica* | 36–37.5 | 7 | 200 | [66] |

*3.2. Factors Affecting the Production of Pasteurella spp.*

The growth substrate, growth supplements, pH, dissolved oxygen (DO), temperature, and incubation time all have an impact on the development of *Pasteurella* spp. by fermentation, which can promote high viability and biomass cells when optimized. Figure 2 depicts the variables that influence the fermentation process of *Pasteurella* spp. after 30 h of incubation time. According to Oslan et al. [9], using growth substrate (yeast extract, glucose, sodium chloride, and sodium phosphate), growth supplements (glutamic acid, cysteine, glycine, methionine, lysine, tyrosine, and histidine), pH (7.4), dissolving oxygen (250 rpm), and temperature (37 °C) can increase by up to 16 times the growth of the gdhA derivative *P. multocida* B:2 mutant.

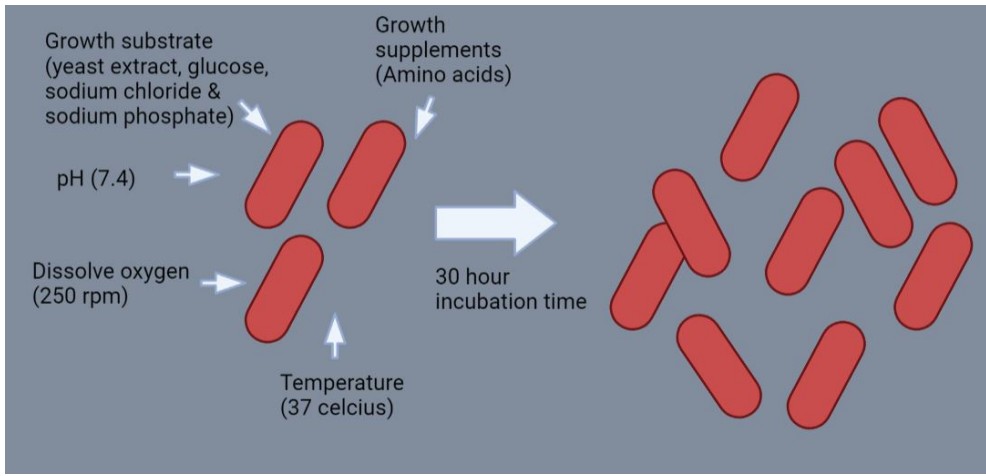

**Figure 2.** The growth conditions of the gdhA derivative *Pasteurella multocida* B:2 in YDB medium [9].

### 3.2.1. Growth Substrate

Growth medium formulation is a significant parameter in the production of high-viability vaccines and bacterial biomass. Nutrients, particularly carbon and nitrogen sources, are of the utmost importance to determine the feasibility of a fermentation process that promotes the increment of biomass and the cell viability of bacteria [54,57,67]. For example, the mutant of the *P. multocida* B:2 vaccine has been grown on a variety of carbon sources, encompassing glucose, sucrose, white sugar, and soluble starch [52]. Notably, the growth and product yield of the *Pasteurella* spp. was dependent on the different types and concentrations of the carbon source, wherein the cell biomass of mutant *P. multocida* B:2 vaccine was elevated when glucose was used as the sole carbon source. Similarly, glucose was identified as the best carbon source and, thus, the preferred growth substrate compared to fructose, d-xylose, galactose, d-mannitol, d-ribose, d-sorbitol, sucrose, lactate, and glycerol in the production of LKT in *M. haemolytica* [68].

The basic function of the carbon source includes the growth rate increment, cell biomass synthesis, and cell maintenance in bacteria [69]. It is noteworthy that glucose is frequently used as the raw material for industrial fermentations because it is relatively inexpensive and provides the most effective carbon source as well as an energy source. However, the formation of by-products by bacterial cells is strongly affected by the composition of the culture medium. When glucose was used as the carbon source, a large amount of acetic acid was synthesized as a metabolic by-product by *M. haemolytica* [51]. The level of acetate accumulation in *E. coli* cultures was different when different carbon sources, including glucose, mannose, and fructose, were tested [70]. Previously, the production of human vaccine HPV16 L1 protein using yeast cultures from *Saccharomyces cerevisiae* significantly jeopardized the yield and the quality of the produced vaccine when the different concentrations and compositions of the carbon source were tested [71,72].

Nitrogen is an absolute requirement for cell growth and is assimilated by the synthesis of glutamine and glutamate. In a previous study, different nitrogen sources with several organic and inorganic sources at different concentrations were demonstrated by Oslan [52] to increase the biomass of mutant *P. multocida* B:2, such as yeast extract, peptone, ammonium chloride, and ammonium sulfate. Interestingly, the number of viable cells and bacterial biomass increased when the nitrogen content in the medium was increased. Furthermore, the organic nitrogen source of the yeast extract was the most important factor in biomass cultivation since it significantly improves biomass production in terms of the growth of *P. multocida*. Moreover, van Rensburg and du Preez [62] have found that yeast extract appeared to be the main compound by which to satisfy the nutritional requirements of *M. haemolytica*; thus, it was unnecessary to perform cultivation in the rich medium of BHI broth to achieve high LKT levels.

Importantly, yeast extract has frequently been utilized as a nutrient ingredient in most fermentation studies, since it provides the micronutrients needed for microbial growth [67]. It has recently been shown to present the best organic nitrogen source for cell growth as it is rich in different amino acids, vitamins, and other growth-stimulating compounds [73]. In addition, Cheng et al. [70] have reported that yeast extract and peptone are rich in free amino acids that support cell growth, with a low consumption rate of the carbon source, and minimize the accumulation of by-products in the substrate feeding strategy to increase the cell density of *Streptococcus suis*. In addition, peptone has also been reported as a good nitrogen source in obtaining a high dry weight of *Pasteurella* spp. [52].

On the other hand, soya casein with amino acids and vitamins has been proven to be a good medium content that supports growth for the in vitro cultivation of *Pasteurella* spp. [74]. Notably, Sarwar et al. [61] and Mehmood et al. [10] have investigated the effect of the medium on the growth of *P. multocida* and its biomass production, using casein sucrose yeast (CSY), tryptic soy (TS), BHI, and nutrient broths. The results have reported a positive correlation between growth rate and the dry mass of bacteria. Multiple studies have routinely used BHI broth to study the biomass production of *P. multocida* for HS



vaccine preparation [53,60]; however, this is cost-ineffective on an industrial scale and is less feasible for commercialization.

### 3.2.2. Growth Supplements

Generally, vitamins are essential as coenzymes for certain enzymes in functional groups, while trace amounts of amino acids are necessary for the synthesis of proteins in bacterial cells [4]. Furthermore, Oslan et al. [9] have studied the effect of various amino acids (cysteine, glutamic acid, glycine, histidine, tyrosine, lysine, and methionine) and vitamins (biotin, riboflavin and vitamin B1, pantothenic acid, pyridoxine, and nicotinic acid) on the growth performance of *P. multocida* B:2. Significantly, histidine was found to greatly enhance the growth of mutant *P. multocida* B:2 by about 19 times. Thus, histidine amino acid is essential as a source of carbon, nitrogen, and energy supply [9]. Among the vitamins tested, riboflavin presented the highest impact on the viability of the mutants' colonies. However, the combination of histidine and riboflavin at certain concentrations (20 mM histidine and 0.02 mg/L riboflavin) in the culture did not promote cell growth or cell viability.

According to Oppermann et al. [68], the optimal growth culture for *M. haemolytica* comprised nicotinamide, calcium pantothenate, and the pyrophosphate of thiamine. However, a higher concentration of vitamins did not affect the increase in LKT expression [51]. Similarly, several other nutrient components, such as pyridoxine, biotin, riboflavin, oleic acid, *p*-aminobenzoic acid, hemin, and folic acid, did not affect the growth of *M. haemolytica* [68,75]. In their study, van Rensburg and du Preez [62] have demonstrated that the cultivation batch of *M. haemolytica*, when supplemented with amino acids, ferric iron, manganese, cysteine, and glutamine, resulted in the rise of the specific rate and volumetrics of LKT production.

For *M. haemolytica*, cysteine is essential for its metabolism, since this species cannot reduce sulfate for incorporation into sulfur-containing amino acids as a nitrogen source [51,76]. Moreover, du Preez et al. [51] revealed that the concentration, yield, and specific production rate of LKT by *M. haemolytica* can be improved through the supplementation of amino acids (cysteine and glutamine) and other vitamins. Alanine and isoleucine are essential for rapid cell growth in *M. haemolytica*, while methionine demonstrates a small inhibitory effect on cell growth, whereas hydroxyproline, serine, and either tryptophan or threonine are totally dispensable for bacterial growth [68].

Nevertheless, the phosphorus that is used as a buffer to resist any change in neutral pH throughout the fermentation process is essential in nutrient-enriched media [9]. It is also crucial for bacterial growth rate maintenance and cell repair. Furthermore, phosphorus could be used to synthesize proteins, carbohydrates, and fats in bacterial cells. The optimized concentration of phosphorus is needed because a reverse effect on product yield could possibly occur [52]. The addition of sodium chloride (NaCl) is important for controlling the increasing temperature in the medium, since a high temperature may cause stress responses in the bacterial cell cultures. Therefore, NaCl is essential for enhanced biomass production in a non-toxic and sustainable medium [9]. Passanha et al. [77] also demonstrated a higher yield (65–77% (*w/w*)) and more cell biomass (6% (*w/w*)) when NaCl was added to the nutrient media. To our knowledge, and from a literature search, reports on the effects of amino acids, vitamins, phosphorus, and salt (NaCl), particularly on the growth of other *Pasteurella* spp., were limited. Hence, supplementation to enable improved media formulation should be explored and optimized, to further enhance the yield and cell biomass of *Pasteurella* spp. for vaccine production.

### 3.2.3. pH

Bacterial enzymatic activity is greatly affected by the pH value of the extracellular environment. The excellent growth of *Pasteurella* usually occurs at a pH ranging from 6.0 to 8.0 [53]. However, *P. multocida* failed to grow at pH 2.0 (acidic) and pH 10 (basic). The optimum pH is generally dependent on the type of medium used for each culture. For

instance, different combinations of sugars in the media have been shown to significantly affect the pH during cell growth, resulting in the maximal viability of *P. haemolytica* and *P. multocida* [65].

In the meantime, *P. haemolytica* has been reported to reach its highest growth rate when the initial pH of the mixed media was within the range of 7.2–7.8 [68]. The growth started to slow down and even completely attenuate when the pH value was less than 6.5 or more than 8.8. In addition, when at a neutral pH, the medium contributes to higher cell biomass and viability of *Pasteurella* spp. than when it is used in acidic and basic environments [4]. Notably, Oslan et al. [78] revealed the lower cellular activity of enzymes in a mutant of *Pasteurella* spp. when the fermentation process was conducted at a higher pH, where the cell biomass and growth rate were also impacted. Hence, the effect of pH on biomass and viable cell production is completely dependent on the composition of the medium.

### 3.2.4. Temperature

Rapid microbial growth can only be achieved within the range of the optimum temperature. In general, the optimum temperature for the growth of *Pasteurella* spp. is between 35 °C and 37 °C [32]. According to Shah et al. [53], *P. multocida* survives at 35–40 °C and shows weak activity at a temperature below $25 \pm 5$ °C. However, the microorganism is not able to grow at temperatures higher than 50 °C. Berkman [66] reported that *P. septica* cultivated on amino acid media and gelatin hydrolysate medium grew the best at 37 °C. Moreover, Hills and Spurr [79] investigated the consistency in *P. septica* growth at two cultivation temperatures of 36 °C and 37.5 °C. Their experiments revealed that the growth of *P. septica* at 36 °C was dependent on the amino acid components of the media, with optimal growth achieved on media containing 20 amino acids.

### 3.2.5. Dissolved Oxygen (DO) Demand

Besides temperature and pH, agitation and aeration have also been reported to be extremely influential in the growth of *P. multocida* B:2 [53,61]. Oxygen plays an important role in vital cellular activities by forming adenosine triphosphate (ATP) and producing energy that can be supplied for bacterial culture. Hence, the bio-oxidative respiration process is necessary and the gaseous requirement in most of the cells can be taken at atmospheric oxygen. The dissolved oxygen (DO) functions as a gas carrier in the fermentation medium to promote bacterial biomass production. In addition, the DO control strategy enables better growth for the enhancement of cell yield and the productivity of bacterial cells [61]. Based on a batch-cultivation study of *P. multocida*, the growth culture and the biomass concentration showed a linear correlation that was identified as a growth-association formation. When it grew higher, the level of biomass production also improved [10]. Notably, the production of biomass occurred during the stationary growth phase. It was concluded that the optimal dissolved oxygen tension (DOT) for cell growth plays an important role in retaining normal metabolism, as a terminal electron acceptor in the electron transport chain but is potentially toxic when oversupplied to the bacterial cells [80].

Furthermore, the mutant *P. multocida* B:2 culture remained stable and avirulent at dissolved oxygen levels of between 20 and 70% in a 5 L bioreactor. In addition, the limited amount of oxygen significantly influenced the metabolism of mutants by causing limited growth and biomass production. Cultivation in the bioreactor also showed that bacterial cells responded defensively to oxygen starvation and exposure to high shear stress by producing low biomass under environmental stresses [81]. Notably, the maximum growth of the bacterium was observed when the broth was stirred at 500 to 550 revolutions per minute (rpm) and the cell growth declined at an agitation rate of 600 rpm during the fermentation of *P. multocida* [61]. Several of the general conditions used in the cultivation of *Pasteurella* spp. are listed in Table 4.

**Table 4.** *Pasteurella* cultivations in different modes (batch and continuous) in bioreactor operation.

| Strain | Mode | Improvement | References |
|---|---|---|---|
| *gdhA* derivative *P. multocida* B:2 | Batch | Biomass production increased by 16.6% in bioreactor | [61] |
| *P. multocida* B:2 | Batch | Achieved high biomass production | |
| *P. multocida* B:2 | Batch | Improved biomass production in short duration | [60] |
| *P. multocida* A | Batch | Improved biomass production | [59] |
| *P. multocida* | Batch | Enhanced colony-forming unit (CFU) and dry mass production | [10] |
| *M. haemolytica* | Batch | Achieved high-viability cells in production | [64] |
| *M. haemolytica* | Continuous | Leukotoxin production was increased by 45-fold | [62] |
| *M. haemolytica* | Continuous | Leukotoxin production was maximized and acetic acid production was minimized | [51] |

## 4. Conclusions

*Pasteurella*-based vaccines are crucial for reducing the huge economic and domestic wildlife losses resulting from *pasteurellosis*. Its conventional mass production and manufacturing processes, which use the less cost-effective BHI broth, should be improved for vaccine commercialization. Hence, the importance of physicochemical factors (growth substrates, supplements, pH, temperature, and dissolved oxygen demand) in enhancing the growth density of cultivated *Pasteurella* spp. is covered in this review. Further optimizations on the bioprocess requirements of *P. multocida* and *M. haemolytica*, including biomass, cell viability, and LKT production, are undeniably significant for vaccine manufacturing. However, strain-specific optimization remains crucial, although the optimized parameters can potentially be inferred from other commonly equipped strains. Researchers in the field can therefore determine the optimized parameters of physicochemical factors at a faster pace by narrowing down the parameter selections and concentration range via reference to the previous successful optimization studies reviewed in this study.

In the case of *pasteurellosis* vaccines, the efficacy achieved in controlled vaccine trials may not necessarily reflect real-world situations. Some of these critical issues and limitations may be exacerbated by regulatory obstacles. Most of these issues may be insignificant if vaccine design is performed with great care and via trial and error. First, the method of administration of the challenge (e.g., IP injection) is not comparable with the spread of a natural infection. Second, very few studies have included field trials, such as those conducted on actual agricultural farms. Overall, despite the fact that some formulations have shown promising results and clear potential, more research and larger-scale trials are required before the described experimentally developed vaccines can be commercialized.

**Author Contributions:** Conceptualization, S.N.H.O. and S.N.O.; writing—original draft preparation, S.N.H.O. and J.S.T.; writing—review and editing, S.N.H.O., S.N.O., A.H.Y., A.Z.S., M.A.A., A.M.L., S.J.L., M.Z.S. and A.B.A.; supervision, A.B.A. All authors have read and agreed to the published version of the manuscript.

**Funding:** The APC was funded by Research Management Center, Universiti Malaysia Sabah.

**Institutional Review Board Statement:** Not applicable.

**Informed Consent Statement:** Not applicable.

**Acknowledgments:** The authors acknowledge financial support from the Research Management Center, Universiti Malaysia Sabah, for the publication fee funding.

**Conflicts of Interest:** The authors declare no conflict of interest.

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
