# Peer review of "Pasteurellosis Vaccine Commercialization: Physiochemical Factors for Optimum Production"

_processes, doi:10.3390/pr10071248_

Round 1

Reviewer 1 Report

Thank you for considering me for reviewing the manuscript;

“Pasteurellosis Vaccines Commercialization: Physiochemical Factors for Optimum Production”

In this multiinstutional review, Oslan and colleagues reviewed the strategies used for manufacturing the vaccine against pasteurellosis. In addition, they reviewed the physicochemical factors related to process conditions that maximise the production yields. This review will help researchers in the field optimise the physicochemical factors used for growing P. multocida and M. haemolytica to help produce vaccines and reduce the economic and animal loss resulting from pasteurellosis.

The review has presented the relevant results quite well and is presented in a well-structured manner. It is an important addition to the literature. However, some points have to be addressed and considered in a revised version:

  • Page 2, last paragraph, line 89: Should be replaced by haemolytica is a member of the family Pasteurellaceae (formerly Pasteurella haemolytica biotype A). Similar issue on page 10, paragraph 3, line 369.
  • Page 3, line 101: What about using MALD-TOF in diagnosis?
  • Page 3, line 112: The sentence is not related to this paragraph; consider removing it—similar issue with figure 1.
  • Page 4, line 142: Please add the full name of the abbreviation OMPs.
  • Page 5, line 168: Do you refer to a specific vaccination program?
  • Page 9, line 329: Reference is required.
  • Table 1: haemolytica should be replaced by M. haemolytica.
  • Table 2: Abbreviations used in the table need to be present in the footnote.
  • Throughout the text, the authors used the words “ to date, nowadays, recently” and did not refer to recent publications. Please consider changing the words.
  • Conclusions: Please consider adding limitations and future directions.
  • References: Some references are missing page numbers.

Author Response

Corrections have been made in accordance with the reviewer's recommendations. Please see the attachment.

Reviewer 2 Report

Pasteurellosis causes considerable animal loss and economic damage around the world every year. In the current review article, the authors have addressed this problem and discussed about the recent progress in developing Pasteurella-based vaccines-a promising way of preventing the spread of the Pasteurella spp. bacteria. The authors have recognized the cost-ineffectiveness of the traditional manufacturing process using BHI broth and discussed different parameters associated to the production. The authors have discussed different physicochemical factors like growth substrates, supplements, pH, temperature, and dissolved oxygen demand that are crucial to the effective vaccine production and cost efficiency. In this context, the article has high importance to the respective field of research and deserves to be considered for publication. Hence, I recommend the article to be published after the minor revision mentioned bellow:

  1. Please provide literature references for the statement made between the lines 48 and 52.

Author Response

(The authors gave the same response as above.)
